# Clinical and Regulatory Concerns of Biosimilars: A Review of Literature

**DOI:** 10.3390/ijerph17165800

**Published:** 2020-08-11

**Authors:** Vesa Halimi, Armond Daci, Katerina Ancevska Netkovska, Ljubica Suturkova, Zaheer-Ud-Din Babar, Aleksandra Grozdanova

**Affiliations:** 1Faculty of Pharmacy, University Ss. Cyril and Methodius, 1000 Skopje, North Macedonia; vesaahalimi@gmail.com (V.H.); kaan@ff.ukim.edu.mk (K.A.N.); ljsu@ff.ukim.edu.mk (L.S.); 2Department of Pharmacy, Faculty of Medicine, University of Prishtina, 10000 Prishtina, Kosovo; armond.daci@uni-pr.edu; 3Department of Pharmacy, University of Huddersfield, Huddersfield, West Yorkshire HD1 3DH, UK; Z.Babar@hud.ac.uk

**Keywords:** biosimilars, clinical practice, interchangeability, extrapolation, pharmacovigilance, regulatory, knowledge, confidence, education

## Abstract

Although biosimilars have been part of clinical practice for more than a decade, healthcare professionals (HCPs) do not fully accept them. This is because of the perception that biosimilars may not be like their originators in terms of quality, safety, and efficacy. This study aims to evaluate the current knowledge and attitudes of healthcare professionals toward biosimilar prescription, and to elaborate on their concerns. We reviewed the literature using PubMed, Cochrane Library, and Science Direct electronic databases in the period from 2018 to 2020. The knowledge and confidence of healthcare professionals vary between countries, between clinical profiles and between studies. Although most of the healthcare professionals had a positive attitude to prescribing biosimilars, they would still prefer to prescribe them in initial treatment. Generally, HCPs were against multiple switches and substitution of biosimilars at the pharmacy level. HCP’s key concern was interchangeability, with eventual consequences on the clinical outcome of patients. HCPs still approach biosimilars with caution and stigma. HCPs need to have an unbiased coherent understanding of biosimilars at clinical, molecular and regulatory levels. It was also observed that most of their concerns are more theoretical than science-based. Physicians are in an excellent position to accept biosimilars, but they need the additional support of regulatory authorities to approve and take into consideration the available scientific data regarding biosimilars.

## 1. Introduction

Biological medicines or biologics have shaped modern medicine by drastically changing the prognosis for many severe and life-threatening diseases such as cancers, diabetes and autoimmune diseases (rheumatoid arthritis, Morbus Crohn, multiple sclerosis, severe psoriasis) and rare diseases [1,2]. Biologics are fundamentally distinct from conventional medicines in terms of nature or origin, structural complexity and variability, manufacturing process, side effects (immunogenicity), formulation, sensitivity, and regulatory aspects [1]. But the crucial obstacle for accessing biologic medicines is their high cost, due to their lengthy and risky development process [1,2]. Originator biologics are novel medicines that consist of active substances made from living cells or organisms and are manufactured through biotechnology, using complex system cells and recombinant DNA technology [1].

To award research and innovation, but at the same time to create the opportunity for market competition and access to therapies, novel biologics enjoy two mechanisms of protection: patents (which usually last up to 20 years), and a period of data exclusivity and market exclusivity (for up to 11–12 years) [3]. Hence, it does not necessarily mean that an already approved biosimilar can immediately seek the market [2,3]. A biosimilar is released on the market and can be available to clinics only after its originator biologic reaches expiry of all patents, which sometimes can be issued after the originator is already in use by patients [4]. Regardless of various definitions, terminology and regulatory approaches for investigating and approving biosimilars, biosimilars can be defined as non-identical copies of originator biologics, determined to be of similar quality, safety and efficacy to them [1,5,6]. Since the biosimilar approval pathway has been abbreviated, biosimilars have become available to patients at lower cost [2]. Therefore, biosimilars have the potential to reduce overall medicine expenditures, but most importantly to increase access to biologic therapies, thereby improving patient outcomes [2,7]. The cost should not be the primary decision for healthcare professionals, and it is within their responsibility to prescribe safe and effective therapies of the required quality.

By leading on the establishment of general and product-specific guidelines for the development of biosimilars, updated over time, the European Medicines Agency (EMA) has approved the highest number of biosimilars to diverse active substances [8]. Since 2006, the EMA has approved six biosimilars of trastuzumab, two of bevacizumab, four of infliximab, 11 of adalimumab (two of these were withdrawn from the market, at the request of the manufacturer), seven of rituximab, three of etanercept, nine of filgrastim (two were withdrawn from the market at the request of the manufacturer), seven of pegfilgrastim, two of enoxaparin, three of epoetin alfas, two of epoetin zeta, one of insulin lispro, one of insulin apart, three of insulin glargines (one withdrawn), two of teriparatide, two of follitropin alfas, and two of somatropin (one withdrawn from the market at the request of the manufacturer) [9]. In contrast to the EU, nearly a decade later, through the Biologic Price Competition and Innovation Act of 2009, the US Food and Drug Administration (FDA) has approved the first biosimilar Zarxio (Filgrastim-sndz) in 2015, and to date, we have not seen the issuance of all product-specific guidelines [10]. Up to now, the FDA has approved five trastuzumabs, two bevacizumabs, four infliximabs, six adalimumabs, two rituximabs, two etanercepts, two filgastrims, four pegfilgrastims, and one epoetin alfa [11]. The World Health Organization (WHO), by issuing several guidelines for biosimilars (or as they call them “similar bio therapeutic products”), keeps trying to align regulatory aspects for developing and approving biosimilars across countries [6,12,13,14]. In 2019 there are 95 approved biosimilars in use worldwide [2].

Medical school, where healthcare professionals initially acquired knowledge on the principles of therapy, do not extensively include biologic medicines as part of their curriculum [7,15]. Therefore, many researchers have suggested that educating healthcare professionals who are involved in prescribing and dispensing biologics and biosimilars will play an important role in the acceptance of biosimilars in clinical practice [7,15,16,17]. Having regard of the fact that each country has its policies for purchasing and pricing of biosimilars, this review principally aims to explore the current knowledge, attitudes, and awareness of healthcare professionals involved in prescription of biosimilars between 2018–2020. It also further extends the attempt to converge the regulatory, clinical and scientific aspects of biosimilars.

## 2. Materials and Methods

### 2.1. Search Strategy

To conduct and explore this review, we mainly followed the PRISMA guidelines for systematic reviews [18], although “state of the art” methodology was also used. To identify relevant studies, we used the following database: PubMed, Cochrane Library and Science Direct between 2018–2020. To explore biosimilars in clinical practice, we used the following keywords: “biosimilar”, “biologics”, “biosimilars”, “follow on biologics”, “biologics, subsequent entry”, “subsequent entry biologics”, “knowledge”, “practice”, “perception”, “awareness”, “questionnaire”, “survey”. We combined keywords with Boolean connectors and the search model was adapted according to each database software. For our second aim, we used reliable regulatory information retrieved from the European Medicines Agency (EMA) [19], the Food and Drug Administration (FDA) [11], and the World Health Organization (WHO) [20]. Several peer-reviewed articles (37–65), retrieved following the snowball citation, were also included.

### 2.2. Study Selection

The initial studies retrieved from the databases were first selected, and studies that met the eligibility criteria, were reviewed and analyzed. Inclusion and exclusion criteria are given in Table 1. By applying these criteria, from 407 studies, 16 were included in the analysis (Figure 1). No studies were excluded because of poor quality. Although we analyzed studies between 2018–2020, two systematic reviews were included which were conducted between 2014–2020.

### 2.3. Data Collection Process

Each study that met the inclusion criteria was analyzed from several aspects such as authors, date of publication, time when the study was conducted, study design, objectives, results, and limitations.

## 3. Results

After screening for eligibility, we included 16 studies in the analysis (Table 2) [21,22,23,24,25,26,27,28,29,30,31,32,33,34,35,36]. The included studies were from Europe, UK, United States, Australia, Asia, and Africa [21,22,23,24,25,26,27,28,29,30,31,32,33,34,35,36]. The targeted healthcare professionals were: clinicians, GPs, pharmacists, nurses, consultants, care managers, and specialists in clinical settings where biologics are more involved such as oncology, rheumatology, endocrinology, gastroenterology, dermatology, nephrology, and hematology. We included four studies which originated from multiple countries. By analyzing the current knowledge, attitudes and perceptions of healthcare professionals regarding biosimilars, we categorized clinical and regulatory concerns about biosimilars in three major branches: interchangeability, extrapolation and pharmacovigilance reporting (Table 3). The identified concerns can be justified from the fact that gaps still exist in understanding the fundamental concepts of biologics and biosimilars across studies. The need to comprehend the development, regulatory approval, extrapolation, interchangeability, and post-marketing surveillance of biosimilars is still evident among healthcare professionals.

### 3.1. Current Knowledge and Attitudes of Healthcare Professionals toward Biosimilar Prescription

In seven studies [21,22,27,29,31,33,34], we found that healthcare professionals had a good knowledge of biosimilars, and at least know the basic concepts of them. However, healthcare professionals were not satisfactorily familiar with concepts and the processes of interchangeability, extrapolation and pharmacovigilance. Nevertheless, the knowledge and awareness of healthcare professionals vary between countries, studies, and clinical profiles. For instance, in Aladul et al.’s study, consultants, nurses and pharmacists had good knowledge of biosimilars and were keen to use them in initial treatment [21]. Giuliani et al. found that although 79.2% of oncology prescribers rate their general knowledge of biosimilars as average to high, only 36.3% of them correctly selected the questions regarding interchangeability [22]. By systematically analyzing 17 studies from Europe and three from the US, Leonard et al. found an overall lack of biosimilar knowledge and low prescribing comfort [23]. The study from Latin America reports a lack of awareness among rheumatologists about the availability of biosimilars, automatic substitution and the understanding of the nomenclature of biosimilars [24]. The Russian study found that only 20% of clinicians considered that biosimilars are not equal to generics [25]. However, Teeple et al. reported that 88% of rheumatologists, dermatologists and gastroenterologists knew what biosimilars are [27]. The study from Spain found that only 27% of primary care physicians know the difference between generics and biosimilars, and 84% of them do not understand biosimilar clinical development [28]. In Aladul et al.’s survey of 2019, 6% of healthcare professionals (mainly nurses) had never heard of biosimilars [29].

Cook et al. reported that 74% of academic oncology clinicians did not know the basic definition of biosimilars and 40.3% of them considered biosimilars and generics as the same molecules [30]. In Park et al.’s survey, 66.2% of Asian gastroenterologists knew the basic concepts of biosimilars, but only 6% of them felt confident to use biosimilar monoclonal antibodies [31]. After receiving printed educational material, the survey of Ismailov and Khasanova found that over 90% of oncology/hematology nurses, nurse practitioners, medical assistants, and patient navigators identified correct answers about the definition, regulation, interchangeability and safety of biosimilars [31]. Sarnola et al. found that 49%–76% of healthcare professionals were familiar with biosimilars, while 2%–25% did not know what biosimilars were [35].

### 3.2. Interchangeability

Most healthcare professionals were not very keen on interchangeability, especially in multiple switches and in automatic substitution. They would prefer to use biosimilars as an initiative therapy rather than switching patients who are already using the originator or switching by the pharmacist without consulting the clinician. HCPs were reluctant about interchangeability because of the possibilities of emerging new adverse events or an increase of prevalence of these. The dominant safety concerns that inhibit healthcare professionals from implementing interchangeability are immunogenicity and allergic reactions. Several healthcare professionals believe that biosimilars do not have the same efficacy profile as their originators. They were also concerned that some biosimilars are not available in the same pharmaceutical dosage as their originators. One barrier that healthcare professionals reported is the use of unique administration devices for biosimilars. Some healthcare professionals would prescribe biosimilars with a trade name, to make sure that the pharmacist would not switch them. To implement interchangeability, most HCPs realized that they need a robust monitoring system and correct engagement in pharmacovigilance records. Even though in some studies healthcare professionals identified the definition of extrapolation, they still lack comprehension of the principles of this scientific rationale.

It is interesting to note that the vast majority of healthcare professionals in rheumatology and diabetology believe that if the cost was equivalent between originator and biosimilar, they would choose the originator biologic over the biosimilar [21]. The survey of Greene et al. that listed 16 strategies for overcoming key barriers to biosimilar adoption found that 84% of managed care and specialty pharmacists agree or strongly agree that US biosimilars are safe and effective for patients when switching [26]. Still, 61% of them had concerns about the safety and efficacy of biosimilars [26].

If the FDA designated the biosimilar as an interchangeable medicine, 94.8% of academic oncology clinicians in Cook et al.’s study would prescribe a biosimilar interchangeably with its originator [30]. In Park et al., 86.7% of Asian gastroenterologists were against automatic substitution at the pharmacy level [31]. In the Polish study, 88% of hospital pharmacists reported concerns that biosimilars are not identical with the originators, 48% with concerns about their immunogenicity and 44% with concerns about other pharmacokinetic properties [33]. In the systematic review of Sarnola et al., 64%–95% of physicians were against the substitution of biosimilars at the pharmacy level [35]. Meanwhile, the Bangladesh study reports that 74% of academicians, 41% of clinicians and 61% of industry experts are positive about interchangeability [36].

## 4. Discussion

The advancement of biotechnology and the recent entry of many novel biologics, biosimilars, bio-betters, and biomimetics may be contributing factors as to why clinicians nowadays don’t approach biosimilars with acuteness. Most clinicians receive continuous education about biologic therapies from professional associations, peer-reviewed articles, conferences, brochures, guidelines from regulatory authorities, pharmaceutical companies, and information from their colleague pharmacists. We still cannot judge whether HCPs receive biased information, but some healthcare professionals perceive biosimilars as therapies with low safety and efficacy.

To provide reliable regulatory and scientific information for healthcare professionals, the European Medicines Agency, jointly with the European Commission and other scientific experts, has established an information guide for healthcare professionals available in 23 languages of the European Union [1]. As patients also need to be considered as part of clinical decisions, and need to know the therapies to which they might have access, the European Medicines Agency together with other experts from relevant fields has established a consensus information leaflet for patients who want to understand biosimilars [37].

So far, all of these regulatory and clinical concerns of HCPs are more theoretical than science-based [38,39,40,41,42,43,44,45,46,47]. Perhaps when biosimilars were introduced, they should have been presented together with their originators, explaining that concerns could not only emerge from biosimilars, but also from their originators among batches. However, despite the education of healthcare professionals, regulatory authorities across countries need to converge guidelines for developing and approving biosimilars [48]. Regardless of the results, the studies had limitations, and some of them did not have a representative power size and response rate, therefore we cannot know exactly if these are the actual views of healthcare professionals regarding biosimilars. Although the studies have been published in the last two years, we must take into consideration that some of them have been conducted earlier. Therefore, these do not necessarily represent the exact reflection of current acquaintance with biosimilars among healthcare professionals.

### 4.1. Biosimilars from a Regulatory Perspective: What Should Healthcare Professionals Know?

Inherent structural variability and the non-identical issue were present in biologics before the existence of their copies known as biosimilars [1,6,49]. This is because of the inability to replicate biological molecules [49]. Therefore, clinicians should take into consideration that no two biologics even with the same active substance and produced by the same manufacturer are identical and perhaps each biologic is different in itself [50].

Moreover, comparability studies of the biosimilar development paradigm and extrapolation are not novel concepts; they have been in existence for some time and they continue to be implemented in originator biologics when there is any change during the manufacturing process [51]. Therefore, originator biologic manufacturers need to demonstrate that the proposed change is not clinically meaningful and does not alter the clinical outcome [51]. Nevertheless, the manufacturer of the proposed biosimilar is not as familiar with the manufacturing process as is the manufacturer of the originator biologic, thus creating intentional copies by trying to reverse the manufacturing process of the originator [50]. Therefore, before developing a new biosimilar, it is crucial to fully comprehend the physical, chemical, biological and microbiological attributes of the originator biologic [1,6,52]. The major responsibility of regulatory authorities and manufacturers is to avoid any clinically meaningful structural difference that might affect negatively the efficacy and safety of the proposed biosimilar [1,6,12]. This is accomplished by assessing and respectively demonstrating a high scale of structural and functional similarities between both [1,10,11]. The biosimilar investigational pathway aims to demonstrate neither superiority nor inferiority between the biosimilar candidate and the originator, but to demonstrate that the biosimilar does not have a decreased or increased safety and efficacy profile compared to the originator [53]. This pathway is distinct from the pathway established for originator biologics and even more distinct from generics [54,55]. Therefore, healthcare professionals should not equate biosimilars with generics (Table 4) [1,6,54,55].

The investigation of biosimilars begins with analytical and functional studies of the biosimilar candidate itself and then with the originator [1,6]. The outcomes from the results of this crucial phase determine the need for additional specific studies in non-human and human primates [1,12,52]. The investigational clinical pathway starts with phase one pharmacokinetics and pharmacodynamics studies (if relevant markers exist) in healthy participants and phase three confirmatory clinical trials between the biosimilar candidate and originator [55,56]. Moreover, at least one immunogenicity clinical trial is required to compare immunogenicity aspects between biosimilar and originator biologic [55,56]. Since clinical trials have already been established for originator biologics, the aim of biosimilar clinical investigation is not to establish the clinical benefit, but to demonstrate the clinical equivalence with the originator biologic [56].

#### 4.1.1. Interchangeability: Switching and Substitution

Currently, interchangeability falls within the scope of clinical practice and it is used to describe the process of transitioning from the originator to the biosimilar and back and forth or between two biosimilars [1,55]. Interchangeability is assessed after the biosimilar gets approval from the regulatory authorities, not simultaneously during the approval [52]. Until now, no biosimilar has been approved to be used interchangeably with its originator biologic, although some biosimilars with low molecular weight like insulins are favorable candidates to obtain interchangeability status soon, and when that happens these therapies will get one year of market exclusivity [57,58]. The EMA does not take responsibility for interchangeability and they leave this with the EU member states [1,55]. Moreover, the FDA requires additional clinical trials which include at least three switches between the biosimilar candidate and the originator biologic [57]. In European countries, several national regulatory authorities support switching during the initiative treatment or with the decision of the prescribing clinician, while substitution is not linearly supported between European states [55,56]. In this scenario, the substitution is automatically implemented by the dispensing pharmacist without consulting the clinician [1,55]. In the US, when the biosimilar is designated to be used interchangeably with the originator biologic through additional clinical studies, the pharmacist can dispense and can authorize the automatic substitution [55,57]. Therefore, the FDA’s concept of interchangeability corresponds to the EU’s concept of substitution.

The main theoretical dilemma of clinicians regarding interchangeability, especially in multiple switches and at the pharmacy level, is the fear of the eventual inducement of immune reaction responses (immunogenicity) and their potential consequences in terms of safety and efficacy [59]. However, it is important to note that immunogenicity might be a consequence of several factors like underlying disease, genetic background, age, immune status, including immunomodulating therapy, dosing schedule and structural and manufacturing structural homology, post-translational modifications, modification of the native protein, formulation, and impurities [1,59,60]. Not just biosimilars but also biologics could have immunogenic properties [59,60]. Many researchers and governments have tried to explore interchangeability, either by inducing comparative clinical studies or literature reviews of all the available clinical studies, and several conclusions have been generated either supporting interchangeability or otherwise [55,56]. Still, confirmatory clinical trials are needed to produce significant results. However, together with clinical trials, we need support from regulatory agencies to achieve clarity in physicians’ view of biosimilars.

#### 4.1.2. Extrapolation

Extrapolation is a rationale scientific principle used to describe the process when the proposed biosimilar receives all the approved indications of the originator, while performing comparative clinical studies of just one or two indications [60,61]. This rationale is acquired in phase three confirmatory clinical trials, although the results from each investigational phase impact the extrapolation of indications [61]. Still, limitations exist if particular indications are under patent protection [61,62]. The integral key to comprehending extrapolation lies in the fact that the structure determines the clinical performance of the medicine [61]. Currently, this concept of transferring the safety and efficacy data from one indication to the other is not very clear to regulators and clinicians. Whether the proposed biosimilar will be able to receive all the other indications from the originator, while performing a clinical trial for one indication, depends on several factors: mechanism of action, posology, pharmacokinetics, and population type [61,62].

#### 4.1.3. Post-Approval Surveillance of Biosimilars

The clinical studies required for the approval of biosimilars are often conducted in a small study population, hence these trials are unable to reveal all potential adverse events, particularly rare and delayed ones [1,62]. Therefore, robust post-approval surveillance remains the crucial part of identification, evaluation and prohibiting adverse effects and other questionable concerns, such as elucidating interchangeability [1,63]. The use of distinguishable names for biosimilars is controversial as to whether it would benefit pharmacovigilance would decrease the confidence of clinicians in thinking that the biosimilar has a different clinical profile compared with the originator biologic [62,63,64,65]. The correct engagement of health care professionals in surveillance records and reports can also be considered an important contribution to safety and other related problems considering biosimilars [63,64]. Serialization of biosimilars is also imperative, detecting eventual safety and quality concerns in each package within the same batch, having regard also to the fact that biosimilars and originator biologics are very sensitive to light, high temperature, and could be easily prone to microbial and viral contamination [1,63]. Even a minor change in the manufacturing process or equipment, product handling and container closure, such as regarding the syringe and needle, could impact the quality, efficacy and safety of biologic medicines [6,52,64].

## 5. Conclusions

This study suggests that efforts and incentives should be made to create evidence-based continuing educational programs for biologic therapies. This would help improve understanding of biosimilars not only at the clinical level but also at molecular and regulatory levels. Regardless of government incentives for implementing prescription quotas for clinicians, central regulatory authorities formally responsible for their quality, safety and efficacy should also take several positions. These include the following: (1) to shift the biosimilar development paradigm in order to demonstrate interchangeability when applying for approval; (2) interchangeability should be granted automatically upon approval; (3) to release a position statement strongly supporting strongly interchangeability as a first-course initiative; (4) to harmonize the basic concepts of interchangeability, switching and substitution. Even though it is within the responsibility of regulatory authorities to create a balanced environment between innovation and research (originator biologic) and access to biologics and market competition (biosimilars), its position is imperative in accepting biosimilars in clinical practice. Biosimilars will continue competing with biologics, but without adjustment of these positions we will not be able to realize the full benefits of biosimilars.

## Figures and Tables

**Figure 1 ijerph-17-05800-f001:**
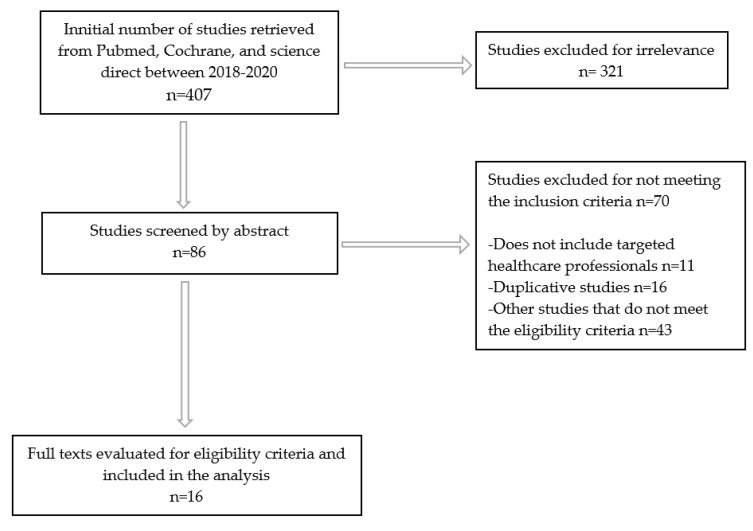
Flow chart of the inclusion and exclusion criteria.

**Table 1 ijerph-17-05800-t001:** Inclusion and exclusion criteria.

Inclusion and Exclusion Criteria	
**Category**	**Criteria**
Year of publication	2018–2020
Geography	No study was excluded because of geography; Studies from Europe, United States, Asia, Australia, UK, Africa.
Language	English and Spanish
Sources	Only peer-reviewed literature
Study design	**First objective**: surveys, interviews and systematic reviews concerning healthcare professionals such as clinicians, pharmacists, nurses, consultants. Narrative reviews, commentaries, editorials, and protocols were excluded. **Second objective**: Reliable information from regulatory agencies such as European Medicines Agency (EMA), Food and Drug Administration (FDA) and World Health Organization (WHO); peer-reviewed articles following snowball citation (37–65).

**Table 2 ijerph-17-05800-t002:** Overview of studies included in the review.

Authors	Overview of Study	Objective	Outcomes	Concerns and Gaps	Limitations	Country
**1. Aladul et al. 2018** [21] Study induced between June–November 2017;	Methods: 30 min face-to-face, semi structured interviews; Sample size: n = 22; Sample: consultants, nurses, pharmacists; Profile: gastroenterology, rheumatology, diabetology;	“To investigate healthcare professionals’ perceptions and perspectives towards biosimilar infliximab, etanercept and insulin glargine and the potential barriers and facilitators to their prescribing”.	Good level of knowledge, and likely to initiate newly diagnosed patient on biosimilars, disagreement with automatic substitution of biosimilars at the pharmacy level and disagreement with multiple switching for cost reasons.	Safety and efficacy concerns (interchangeability, extrapolation), the use of different excipients, and different administration device, unavailability of all dosage strengths of the biosimilars.	Small sample size, diversity of specialties and organizational background, only four pharmacists responded to this interview.	UK
**2. Giuliani et al. 2018** [22] Study induced between September 2017–October 2017	Methods: a 19-question survey; Sample size: Europe (n = 321), Asia (n = 84), US (n = 55), Africa (n = 13), Australia (n = 7); Sample: prescribers; Profile: oncology;	“To assess the current level of knowledge, understanding and comfort of use of biosimilars among prescribers”.	Most prescribers (79.2%) rate their general knowledge of biosimilars as average to high. 74.6% of prescribers were able to identify the most appropriate definition of biosimilars. 57.4% feel comfortable using an EMA-approved biosimilar. Only 62.3% understand extrapolation. 36.3% were able to identify the concept of interchangeability.	Safety concerns (interchangeability)	No hypothesis was tested, only participants of the ESMO (European Society for Medical Oncology) were included and not all responded completely.	Multicentered
**3. Leonard et al. 2019** [23] Studies included between 1 January 2014–5 March 2018	Methods: systematic review; Sample size: US (n = 3) and EU (n = 17); Sample: clinicians, pharmacists, specialty physicians, nurses. Profile: rheumatology, dermatology, gastroenterology, diabetology;	“To evaluate current U.S. and European health care provider knowledge, perceptions, and prescribing behaviors of biosimilar medicines, to assess the need for clinician-directed biosimilar education”.	Overall lack of biosimilar knowledge and awareness, biosimilars mostly used in initiative treatment.	Safety and efficacy concerns, immunogenicity (interchangeability, extrapolation).	Potential for biased interpretation of results. Limitations from the included individual studies.	US, EU
**4. Hernández et al. 2018** [24] Study induced between 6 September–8 September 2017	Methods: short survey comprising six questions. Sample size: n = 104. Sample: clinicians; Profile: rheumatology;	“To determine awareness of biosimilars, including prescribing practices, nomenclature, automatic substitution and ADR reporting”.	Lack of awareness considering availability of biosimilars, automatic substitution and nomenclature.	Not applicable	Not sufficient data considering methodology.	Latin America
**5. Karateev et al. 2019** [25] Study conducted from 15 June–22 July 2016	Methods: survey comprising 15 questions; Sample size: n = 206; Sample: clinicians; Profile: rheumatologists, gastroenterologists, hematologists, oncologists;	“To assess levels of knowledge and attitudes towards biosimilars and key policies on their use among Russian physicians, and to define the level of interest in new information on biosimilars, and determine what evidence drives treatment decisions in Russia”.	80% of respondents lacked understanding of the differences between biosimilars and generics. 67% supported prescribing biologics by distinguishable names. 20% of respondents twice confirmed that biosimilars were different from generics and that they were not identical copies of the originator. 53% were against automatic substitution.	Safety and efficacy concerns (interchangeability).	Study findings are related to local biosimilars in Russia, and does not reflect the opinion of clinicians for international biosimilars.	Russia
**6. Greene et al. 2019** [26] Study conducted from 1–19 October 2018	Methods: Survey comprising 16 strategies for overcoming barriers on 5 point scale; Sample size: n = 300; Sample: managed care and specialty pharmacists;	“To assess perceptions regarding strategies for overcoming barriers to biosimilar adoption among managed care and specialty pharmacy professionals”.	84% of respondents agreed or strongly agreed that FDA-approved biosimilars are safe and effective for patients who switch from a reference biologic. 54% agreed or strongly agreed with extrapolation.	(61%) Safety and efficacy concerns (interchangeability, extrapolation)	First 300 respondents were selected for analysis; potential for biased evaluation.	US
**7. Teeple et al. 2019** [27] Study conducted between June 2016–January 2017	Methods: 15-min online survey; Sample size: n = 297; Sample: clinicians; Profile: rheumatology, dermatology and gastroenterology;	“To understand the level of familiarity of clinicians with biosimilars, their experience with non-medical switching (switching medications for reasons unrelated to patient health) of patients between biologics and their attitudes towards switching from a biologic to a biosimilar”.	88% of respondents knew the definitions of biosimilar, 84% of respondents did not agree to switch stable patients from biosimilars to originators. Only 17% of respondents would feel confident with substitution of biosimilars at pharmacy level. 50% are comfortable with extrapolation.	Concerns about interchangeability, safety and efficacy, immunogenicity, patient mental health, physician office management.	Clinicians were recruited before survey, the outcomes cannot reflect the opinion of clinicians who were not recruited.	US
**8. Micó-Pérez et al. 2018** [28] Study conducted from 19 May 2016–11 September 2016	Methods: questionnaires of 34 questions; Sample size n = 701; Sample: physicians; Profile: primary care;	“To evaluate the awareness and training needs on biosimilars”.	42% of respondents knew the definition of biosimilar, 27% knew the difference between generics and biosimilars, 84% did not understand the biosimilar clinical development.	Not applicable	Not applicable	Spain
**9. Aladul et al. 2019** [29] Study conducted between August 2016–January 2017	Methods: anonymized, self-administered web-based survey. Sample size n = 234; Sample: medical consultants, registrars, pharmacists, nurses; Profile: dermatology, diabetology, gastroenterology or rheumatology;	“To investigate knowledge and attitudes of different healthcare professionals in UK towards infliximab and insulin glargine biosimilars”.	76% of medical consultants/ registrars, 84% of pharmacists knew of the basic concepts of biosimilars. 6% of HCPs (mainly nurses)) had never heard of biosimilars, more comfortable with initiation of biosimilars than switching from the already in use originator.	Safety and efficacy concerns (interchangeability)	A small number of pharmacist respondents 11%,	UK
**10. Cook et al. 2019** [30] Study conducted between January–May 2018	Method: a 12 question survey; Sample size: n = 77; Sample: physicians, pharmacists, advanced practice providers; Profile: oncology;	“To investigate oncology clinicians’ understanding of biosimilars and what information they need prior to adoption”.	74% of respondents didn’t know the basic definition of biosimilars; 40.3% considered biosimilars and generics as same entities. 94.8% of respondents would use an interchangeable biosimilar if it had FDA approval for interchangeability.	Knowledge gaps and the need for education regarding biosimilars is high.	Conducted at a single academic institution.	US
**11. Park et al.****2019** [31] Study conducted between 24 February 2017–26 March 2017	Method: a 17-question multiple-choice anonymous web survey; Sample size n = 151; Sample: clinicians; Profile: gastroenterologists;	“To assess the awareness of biosimilar monoclonal antibodies among Asian physicians”.	66.2% of respondents knew of the basic concepts of biosimilars. Only 19.2% of respondents considered that originator and biosimilars are interchangeable; only 6% felt confident in the use of biosimilar monoclonal antibodies. 86.7% were against automatic substitution at the pharmacy level.	Safety and efficacy concerns (interchangeability, extrapolation, immunogenicity)	Most respondents were only from Korea, Japan, and China.	Asia
**12. Ismailov and Khasanova 2018** [32] Study conducted: Not applicable	Method: a survey of 22 questions; Sample size: n = 62 respondents; Sample: nurse, nurse practitioners‘ medical assistants, patient navigators. Profile: oncology/hematology;	“To increase oncology/hematology team members’ knowledge of biosimilars and then use an anonymous online survey to assess the knowledge gained”.	More than 90% of survey respondents identified correct answers about the definition, regulations, interchangeability, safety, cost issues, and use of biosimilars in oncology.	Knowledge gaps about safety profile, the need for education regarding biosimilars.	Did not discuss complex topics in depth (interchangeability, extrapolation, immunogenicity).	US, Colorado
**13. Pawłowska et al. 2019** [33] Study conducted in September 2017	Method: a paper-based, self-administered questionnaire comprising 12 short questions; Sample size: n = 61; Sample: hospital pharmacists;	“To identify hospital pharmacist opinions towards biosimilars and investigate their usage in practice”.	68% of respondents believed that biosimilars should be used in the initiation of therapy, 75 % of respondents did not agree with the substitution of biosimilars at pharmacy level.	88% of respondents were concerned that biosimilars were not identical with originators, 48% with their immunogenicity and 44% with other pharmacokinetic properties.	The response rate was 22.5% and the results may not be representative of all hospital pharmacists in Poland.	Poland
**14. Hadoussa et al. 2019** [34] Study conducted: Not applicable	Method: anonymous questionnaire comprising 15 multiple-choice questions; Sample size n = 107; Sample: clinicians; Profile: oncology and hematology;	“To evaluate the knowledge and perceptions of Tunisian oncologists and hematologists on biosimilars”.	71% of respondents were able to differentiate the biosimilar from the generic. 11% knew the difference between biosimilars and their originators. About 52.34% of respondents were in favor of justified substitution and interchangeability.	Safety and efficacy concerns, interchangeability, extrapolation.	Not applicable	Tunisia
**15. Sarnola et al. 2020** [35] Studies included between 2014 until the end of 2018	Method: systematic review; Sample size n = 23 studies; Sample: Europe (n = 16); North America (n = 4); Australia (n = 1), New Zealand (n = 1) Central and South America (n = 1); Sample: clinicians; Profile: nephrology, rheumatology, dermatology, neurology, endocrinology, and oncology, gastroenterology;	“To examine physicians’ perceptions of the uptake of biosimilars”.	Physicians’ knowledge and attitudes towards biosimilars vary between studies. 49%–76% were familiar with biosimilars while 2%–25% did not know what biosimilars were. 64%-95% of physicians were against substitution of biosimilars at the pharmacy level.	Safety and efficacy concerns (interchangeability, extrapolation, immunogenicity).	The data extraction from studies was done by only one researcher.	Multicentered
**16. Kabir et al.** 2018 [36] Study conducted: Not applicable	Method: questionnaire-based survey; Sample size n = 250; Sample: clinicians, academics, industry experts;	“To examine whether biosimilars introduced in the Bangladesh drug industry required any further clarification with regard to industrial manufacture, distribution and clinical prescription”.	72% of industry experts, 54% of academics regard biosimilars as drugs with equivalent efficacy as their originators; 41% of clinicians saw biosimilars as bioequivalent with their originators, and no need for clinical trials to be approved. (74%) academics, (41%) clinicians, and (61%) industry experts were positive about interchangeability;	Gaps in knowledge, and harmonization between regulation and science.	Questions regarding the understanding of biosimilars were not clearly formulated by authors.	Bangladesh

**Table 3 ijerph-17-05800-t003:** Major clinical and regulatory concerns of healthcare professionals related to biosimilars.

Clinical and Regulatory Concerns	Definitions
**Interchangeability concepts**	**Europe: Interchangeability: switching and substitution** describes the process of transitioning from the originator to biosimilar and back and forth or between two biosimilars. Interchangeability comprises switching (a transition implemented by the clinician) and automatic substitution (a transition implemented by the pharmacist without consulting the clinician). **United States: Switching** describes the process of transitioning from the originator to biosimilar and back and forth or between two biosimilars at the pharmacy level.
**Extrapolation concept**	A scientific rationale used to describe transferring of the safety and efficacy data from one indication to others, without the need to conduct clinical trials for each indication.
**Pharmacovigilance reports**	Crucial for the identification of adverse events. Using trade name, international nonproprietary name (INN), and batch number is critical, considering that no two biologics even with the same active substance and from the same batch are identical.

**Table 4 ijerph-17-05800-t004:** Regulatory characteristics of biosimilars and generics.

	Biosimilars	Generics
**Originated from**	Biologic medicines	Conventional medicines
**Development paradigm**	Highly similar and not identical to the originator (comparative studies)	Bioequivalent and identical to the originator (bioequivalence studies)
**Approval procedure**	Usually approved from central regulatory authorities EMA or FDA	Usually approved from national regulatory authorities of EU or FDA
**Immunogenicity**	Yes	No
**Nomenclature**	Trade names or the use of distinguishable names with distinct suffixes	INN names
**Interchangeability**	Not yet assessed, established or approved	Yes
**Substitution**	Not yet assessed and established or approved	Yes
**ADR**	Report the INN name manufacturer and batch number	Report the INN name
**Risk management plan**	Yes	No
**Price discount**	20–30% discount over the originator	80–90% discount over the originator
**Timeline development**	8–10 years	3–5 years
**Development cost**	$100–$200 M	$1–$5 M

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
