# Peer review of "Clinical and Regulatory Concerns of Biosimilars: A Review of Literature"

_ijerph, 2020, doi:10.3390/ijerph17165800_

Round 1
Reviewer 1 Report
- Please correct grammar and style throughout manuscript. There were places where the manuscript was difficult to read. There are also a number of places were there are no spaces between sentences.
- The manuscript needs a detailed description of what a biosimilar is and how it is different from other types of drugs on the market. Without that detailed explanation, the importance of the extensive literature search and analysis is lost on the reader. Just from reading your manuscript, I initially did not understand what a biosimilar was or why it was important to warrant this study.
Reviewer 2 Report
The manuscript addresses the important issue of biosimilars. The review describes and analyzes a dozen studies on the perception of genetic drugs and their biosimilars by medical personnel, which made it possible to propose a management strategy aimed at increasing the safety of biosimilars. The researches is described in detail and the conclusions are justified and precise. The manuscript should be published, it does not require a substantive revision, maybe except for the fact that the report by Evelien Moorkens, Arnold G Vulto and Isabelle Huys: Report Biosimilars Regulatory Frameworks for Marketing Authorization of Biosimilars: Where Do We Go From Here? 2018, EPLR, 3, 1-6, DOI: 10.21552 / eplr / 2018/3/6 could be cited.
There are, however, many editorial errors in the manuscript:
Page numbering is not continuous, the numbers start three times with 1.
In Table 2, the 12th study, the place of the research was defined as Poland, Europe. Why was the continent mentioned in case of this country, if it does not appear in any other (Tunisia, Bangladesh, etc.)
Table 4 does not exist, 5 appears after 3.
Spaces are missing in many places, especially between the end of a sentence and the beginning of the next.
Reviewer 3 Report
The review article entitled "Clinical and regulatory concerns of biosimilars: A review of literature" tried to evaluate the current knowledge, attitude and position of healthcare professionals toward biosimilars. I found the article informative and I have only a couple of suggestions:
- Section 3.1.1. and 3.1.2. is a bit difficult to read. I would suggest to reduce this part perhaps by adding pie graphs with the different percentages.
- The english is not bad per se but should be improved in grammar and style.Examples are the use of contractions don't/didn't, which are usually frown upon in written formal english. Perhaps a quick run through an editing service might be the easiest and quickest way to solve the issue.
Author Response
Please see the attachment.

This manuscript is a resubmission of an earlier submission. The following is a list of the peer review reports and author responses from that submission.